# Current Progress, Challenges and Perspectives in the Microalgal-Bacterial Aerobic Granular Sludge Process: A Review

**DOI:** 10.3390/ijerph192113950

**Published:** 2022-10-27

**Authors:** Qianrong Jiang, Honglei Chen, Zeding Fu, Xiaohua Fu, Jiacheng Wang, Yingqi Liang, Hailong Yin, Junbo Yang, Jie Jiang, Xinxin Yang, He Wang, Zhiming Liu, Rongkui Su

**Affiliations:** 1Ecological Environment Management and Assessment Center, Central South University of Forestry and Technology, Changsha 410004, China; 2School of Environmental Science and Engineering, Central South University of Forestry and Technology, Changsha 410004, China; 3South China Institute of Environmental Sciences, Ministry of Ecology and Environment, Guangzhou 510655, China; 4School of Hydraulic and Environmental Engineering, Changsha University of Science & Technology, Changsha 410114, China; 5Department of Biology, Eastern New Mexico University, Portales, NM 88130, USA

**Keywords:** microalgal-bacterial granular sludge, resource recovery, environmental sustainability, wastewater treatment

## Abstract

Traditional wastewater treatment technologies have become increasingly inefficient to meet the needs of low-consumption and sustainable wastewater treatment. Researchers are committed to seeking new wastewater treatment technologies, to reduce the pressure on the environment caused by resource shortages. Recently, a microalgal-bacterial granular sludge (MBGS) technology has attracted widespread attention due to its high efficiency wastewater treatment capacity, low energy consumption, low CO_2_ emissions, potentially high added values, and resource recovery capabilities. This review focused primarily on the following aspects of microalgal-bacterial granular sludge technology: (1) MBGS culture and maintenance operating parameters, (2) MBGS application in different wastewaters, (3) MBGS additional products: biofuels and bioproducts, (4) MBGS energy saving and consumption reduction: greenhouse gas emission reduction, and (5) challenges and prospects. The information in this review will help us better understand the current progress and future direction of the MBGS technology development. It is expected that this review will provide a sound theoretical basis for the practical applications of a MBGS technology in environmentally sustainable wastewater treatment, resource recovery, and system optimization.

## 1. Introduction

Wastewater biological treatment processes include traditional activated sludge technology, aerobic granular sludge technology, biological rotary disk method, biofilm filter, anaerobic fluidized bed, etc. Traditional activated sludge technology is one of the most widely used sewage treatment technologies. Microorganisms metabolize organic pollutants in water under anaerobic and aerobic conditions. However, traditional activated sludge technology generally has problems, such as a poor sludge settling performance, a high energy consumption, a wide area, and an easy sludge expansion, in long-term operations, which seriously restrict the treatment efficiency of wastewater treatment technology [1,2]. In recent years, aerobic granular sludge technology (AGS) has received extensive attention due to its superior settling performance [3], smaller floor space, a high biomass, and high wastewater treatment efficiency, and gradually become a traditional activated sludge technology in wastewater treatment [4,5]. However, the aerobic granular sludge technology still has the common defects of biotechnology. At the wastewater treatment stage, the system stability is greatly affected by the water quality. Slow granulation, a long start-up period and an insufficient long-term operation stability are still the bottleneck of the aerobic granular sludge technology [6,7]. The limitations of the aerobic granular sludge treatment process and the unsustainability of water treatment have become prominent. Thus, it has become increasingly difficult to meet the needs for a low-energy and high-efficiency wastewater treatment. Researchers are seeking a green and sustainable development of the wastewater treatment technology to reduce the pressure on the environment caused by resource shortages [8,9].

Recently, microalgal-bacterial symbiotic aerobic granular sludge (MBGS) technology has attracted widespread attention. MBGS is a composite bioconcentration technology, based on the coupling of microalgae and sludge. MBGS not only combines the high biomass and high treatment efficiency of sludge with a high added value and resource recovery capability of microalgae, but it also solves the problem of a poor settling performance and difficulty in recovery of microalgae, in the water treatment process [10,11,12,13]. At the same time, the microalgae in situ oxygen production and bacteria form a small bioconcentration of oxygen and carbon dioxide cycle, which can use the light source as the only energy source to replace the external aeration with the microalgae oxygen production under light conditions [14,15]. This makes MBGS promising as a new green and sustainable new process to achieve carbon neutrality goals, in future municipal wastewater treatment [16]. At this stage, MBGS is experimented with treating various industrial wastewaters, such as municipal wastewater, farm wastewater, and saline wastewater, and has shown a notable removal capacity. The structure of MBGS is shown in Figure 1. Microalgae are one of the major factors affecting the treatment of wastewater by microalgae and bacterial symbiotic systems [17]. Different species of microalgae are effective in different carrier surfaces and in various types of reactors. Depending on the characteristics of the target wastewater quality, the selection of the appropriate microalgae is critical. So far, a large number of studies have investigated the adhesion capacity, productivity, and wastewater treatment efficiency of different microalgae. Meanwhile, various microalgal species have been utilized for wastewater treatment, such as *filamentous algae* [18], *Chlorella* [19], *S. obliquus* [12], *Chlamydomonas* [20], *Botryococcus braunii* [21], *Nitzschia* sp., and *Cosmarium* sp. [22], *Haematococcus pluvialis* [23], and *Spirulina platensis* [24], and others. Among them, *Chlorella* sp. and *Scenedesmus* are the most frequently utilized [25]. Ji and Wang [10,26] used MBGS to treat municipal wastewater and found that MBGS could achieve a stable growth in wastewater, and had a good removal rate of COD, nitrogen, phosphorus, and other nutrients. At the same time, the system had a good resistance to some toxic substances in wastewater. Cui [27] used MBGS to treat wastewater, enriched with concentrated nutrients, such as nitrogen and phosphorus, from the biomass through microalgae and bacteria, and finally used them in the production of biofeed and biodiesel. MBGS not only possesses a single removal function, but also has the multi-functionality of the synergistic resource-energy recovery and carbon neutralization [11]. The limitations of MBGS for industrial wastewater treatment are mainly the need to combine physical and biochemical means, the generation of greenhouse gases, such as methane, and the poor resource recovery performance. From a green and sustainable perspective, MBGS is a wastewater treatment technology with a great potential, but the long system startup period and the need to improve the stability are still the bottleneck of this technology [28].

The purpose of this review is to summarize the latest research progress of MBGS by focusing on the following aspects: (1) MBGS culture and maintenance operating parameters, (2) MBGS application in different wastewaters, (3) MBGS- related additional products (biofuels and biological products), (4) MBGS energy saving and consumption reduction (greenhouse gas emission reduction), (5) challenges and prospects. The information reviewed in this paper will help us to better understand the general situation of the MBGS operation and its application and resource utilization in water treatment, to clarify the future development direction, and provide theories for the practical application of algal granular sludge in wastewater treatment and system optimization.

## 2. MBGS Culture and Operating Parameters

MBGS is a biological aggregate with a multi-layered structure formed by microalgae and sludge, under characteristic conditions [29]. The formation strategy of MBGS is shown in Figure 2. The formation mechanism of MBGS may provide new insights into rapid granulation. For example, the rational control of the selection pressure helps to control the biomass growth, improve the particle hydrophobicity, and accelerate granulation. The large particle size, compact structure, good sedimentation performance, and unique layered structure of the mature algal aerobic granular sludge, can provide a good living environment for anaerobic bacteria and aerobic bacteria, respectively [30,31]. There are three ways to cultivate MBGS. (1) The first is microalgae + aerobic granular sludge. Specific microalgae, inoculated into aerobic granular sludge, can rapidly form compact biological aggregates. Zhang [15] inoculated microalgae on conventional AGS, and obtained mature algal bacterial sludge particles within 18 days, and the removal rate of the total nitrogen and total phosphorus in wastewater in the MBGS reactor was higher than that of the conventional AGS. (2) The second is microalgae + activated sludge. Microalgae and activated sludge are simultaneously inoculated into the reactor, and agitation or aeration is used to provide shear force to form algal sludge particles [32]. (3) The third is oxygen-containing light particles. Oxygen-containing photogranules, formed under hydrostatic conditions, were inoculated in the reactor, and within two months, the inoculated oxygen-containing photogranules formed mat-like organisms under light conditions, and finally grew into spherical bioaggregates with a multi-layered structure [33]. The reported selective factors affecting the formation of MBGS are: light, shear force, hydraulic retention time, and inoculum concentration [34].

### 2.1. Lighting

In the MBGS system, the participation of light in the photosynthesis of microalgae is inseparable from the biomass growth, particle formation and performance of the system. By controlling the light stability and light intensity, substances, such as signal molecules and proteins, that facilitate particle formation can be induced to synthesis. However, the physicochemical properties and pollutant removal efficiency of microalgal-bacterial sludge particles formed under natural light conditions are not ideal. Huang [35] formed algal sludge granules under natural sunlight irradiation. Compared with aerobic granular sludge, the granulation was slow, the structure was loose, and thus, the removal rate of the nutrient elements was reduced. Because the overgrowth of algae occupies the *anammox* bacteria niche, it is not conducive to the growth of nitrifying bacteria; at the same time, the particle structure is loose, the particle sedimentation is reduced, and the living space of some anaerobic bacteria is affected. The composition of the biological community changed, and the growth of the nitrifying bacteria (especially *Nitrospiriaceae* and *Nitrosomonasaceae*), the denitrifying bacteria, and the phosphorus accumulating bacteria (PAOs), was inhibited.

In the reactor under constant light and dark conditions with a measured light irradiance of 121 ± 7.3 μmol/m^2^, the EPS of the formed particles contained more tryptophan and aromatic hydrophobic proteins, and the activity of the algae-dominated microorganisms was greatly increased. A stable light-dark cycle and light intensity are more conducive to the growth of microalgae, and can induce the particles to produce hydrophobic proteins that are conducive to the formation of a stable algal-bacterial sludge structure, and form algal-bacteria sludge particles with a compact structure and good sedimentation performance. At a low light intensity (142 ± 10 μmol m^−2^ s^−1^), more tryptophan and aromatic proteins, and N-acyl homoserine lactones (AHL) with side chains ≤C_10_ were induced. AHL regulates the EPS production through the inter-organism quorum effect, which in turn promotes the formation of compact algal sludge granules.

### 2.2. Shear Force

The shear force energy provides a certain degree of hydrophobicity, increases the collision and bonding of the particles in the system, and has an important impact on the rapid formation of particles, particle structure, and extracellular polymer production, in the system. Aeration and stirring are commonly used to provide adaptive shear force to the system. Large-sized particles are often difficult to form under strong shear conditions, or large particles are broken and dispersed due to collisions. At a stirring speed of 80 r/min, the particle size formed was smaller than that at a low stirring speed of 50 r/min. Shear force is an important but not necessary condition for the formation of system particles. A light-driven process to produce algal-bacterial symbiotic aerobic granular sludge, is different from the conventional method that uses aeration as shear force. Under light conditions, the inoculation of microalgae and bacteria-containing particles in a static reactor, formed a light mat containing microalgae and bacteria, and later bioaggregates spontaneously formed spherical organisms.

### 2.3. Hydraulic Retention Time (HRT)

Different HRTs can screen microalgae and bacterial populations in the reactor, to varying degrees. A low HRT can screen out the biomass with superior sedimentation. A suitable HRT can discharge hydrophilic bacteria, leaving more hydrophobic bacteria and microalgae, which can accelerate the particle formation [36].

### 2.4. Inoculum Concentration

The seeding density of light particles can affect the formation rate and physicochemical properties of oxygen-containing light particles. A too high or too low seeding density is not conducive to the formation of dense oxygen-containing light particles. Under a low inoculation density, the biomass concentration per unit volume is small, and the biological growth is slow, but, under a high concentration inoculation density, the inoculated light particles have less substrate concentration on average, and some light particles pass endogenous respiration, under the limited nutrient conditions, to sustain their own life activities, resulting in a lower biomass production. When the inoculation density was set at 600, 900, and 1100 mg/L, and the inoculation concentration of 900 mg/L was set in the reactors, the highest extracellular polymer secretion and the earliest compact spherical particles were formed.

### 2.5. Organic Load

The influent organic load affects particle size and distribution and complex organic matter promotes the growth of particles and bacteria. Usually, a higher COD promotes the growth of the biomass, and it is easy to form a granular sludge with a compact structure and large particle size. The particle size formed in the case of influent COD/N = 8, is generally larger than that of influent COD/N = 1. On the 90th day of culture, the particles with a size of 2.0–2.5 mm and >2.5 mm, accounted for 29% and 63%, respectively, in the COD/N = 8 reactor. In the COD/N = 1 reactor, the particles with a size of 2.0 to 2.5 mm gradually dominated.

### 2.6. Algal and Bacterial Species

To treat wastewater more efficiently and effectively, the most suitable algal species must be selected. The selection is mainly based on the ability of the algal species to remove nitrogen and phosphorus and their adaptability to different wastewaters. To amplify the rate of nutrient excretion, algal cells are starved prior to exposure to wastewater, with only a few exceptions, such as *C. vulgaris*, where starvation has negative effects [37,38,39]. The N:P ratio affects the supplement excretion, depending on the included algal species. A decrease in the P removal was only observed in *Klebsormidium* when increasing N:P with a fixed PO_4_ concentration [40,41,42,43,44].

Bacteria have stimulatory and inhibitory effects on the algal growth. The presence of bacteria has a significant impact on the algal biomass formation and nutrient removal. With a limited supply of available nutrients, there is often competition between bacteria and algae. Some bacteria release enzymes, such as *glucosidases*, *chitinases,* and *cellulases*, that break the structure of the water column. Cellulases, released by some bacteria, break the cell walls of algae, which leads to algal lysis. The intracellular compounds of algae are used by the bacteria after the algal cell lysis [45,46,47,48,49].

## 3. MBGS Application in Different Wastewaters

Following the aerobic granular sludge, MBGS has gradually become a biological treatment technology with high expectations, due to its low energy consumption and low CO_2_ emission. At present, MBGS is experimentally used in various industrial wastewaters, such as municipal wastewater, aquaculture wastewater, domestic wastewater, and salt-containing wastewater [50]. The principle of the MBGS treatment of wastewater is shown in Figure 3.

The treatment and operation parameters of MBGS for different types of wastewater are shown in Table 1. Municipal wastewater has been widely concerned because of its strong biodegradability. Ji [26] used algal-bacteria granular sludge, for the first time, to simulate the treatment of synthetic municipal wastewater, and the removal rates of organic matter, ammonia, and phosphorus reached 92.69%, 96.84%, and 87.16%, respectively, within 6 h. Wastewater contains a variety of functional microalgae and bacterial communities, and the removal of organic matter mostly depends on the assimilation function of these microalgae and bacteria. This experiment is a preliminary attempt to use MBGS to treat municipal wastewater. Wang [51] explored the effect of the system for treating municipal wastewater under natural conditions. Studies have shown that the system is affected by changes in ambient temperature and light intensity during the day and night, and the removal rates of COD, nitrogen, and phosphorus also change. Compared to aerobic granular sludge, MBGS maintains good nutrient removal rates from water, while emitting less greenhouse gases [13]. Previously, the removal process of ammonia nitrogen by wastewater biological treatment technology has been well understood and studied, but the removal path of phosphorus is not clear enough. Therefore, Ji [26] explored the mechanism of phosphorus removal under the condition of 6 h light-dark cycle. The study shows that under the condition of 6 h light-dark cycle, the removal efficiency of phosphorus reached 86%, and the accumulation of phosphate was the main route of phosphorus removal in the system. Growth requirements of microalgae and bacteria in the photoreaction stage, phosphorus accumulating bacteria and microalgae quickly absorb a large amount of dissolved inorganic phosphorus in water, and store it in the organism in the form of polyphosphate.

The salinity of the high-salt-concentration wastewater has an inhibitory effect on organisms. Some teams have studied the efficiency of the MBGS treatment of saline wastewater, but the effect of MBGS on wastewater treatment is not ideal under the condition of 4% salinity. Only 17% of the total inorganic nitrogen was removed from the 4% salinity wastewater, the concentration of inorganic phosphorus did not drop but increased, and a large amount of lipids accumulated in the water. Meng [52] studied the feasibility of the MBGS removal of pollutants in the CFR reactors in the salinity range of 1–4%. Studies have shown that in the range of 1–3% salinity, the algal sludge particles maintained a good stability. Under the high (4%) salinity condition, the activities of *Nitrosomonas* and *Flavobacterium* were inhibited, and thus, inorganic nitrogen and phosphorus were not effectively removed.

So far, the MBGS related research has mostly been associated with the treatment of municipal wastewater and salinity wastewater, very little with the treatment of aquaculture wastewater, containing high concentrations of nitrate and nitrite. In 2020, Fan [53] used algal-bacterial granular sludge to treat synthetic aquaculture wastewater in a 60 mL glass reactor. Without aeration, the chemical oxygen demand, ammonia-nitrogen, nitric acid-nitrogen, the removal rates of nitric acid-nitrogen and phosphate-phosphorus were 64.8%, 84.9%, 70.8%, 50.0%, and 84.2%. This shows that MBGS has a great potential in the treatment of aquaculture wastewater.

MBGS has a certain potential to treat low-carbon wastewater. Zhao [54] studied the system stability and nutrient removal efficiency of MBGS in the treatment of low carbon wastewater. Studies have shown that in low carbon wastewater with COD/N = 1, the utilization of algae and bacterial sludge particles for phosphorus is as high as 98%. Huang [55] used a partial nitrification process to establish a partial nitrifying algal bacterial particle system when dealing with low-carbon and low-nitrogen wastewater, which achieved a high-efficiency removal of ammonia nitrogen. The removal efficiency of ammonia nitrogen was 99%, but the accumulation of nitrite also reached 96.5%.

MBGS also shows a great potential in the treatment of other special wastewater. Cai [56] found that when treating wastewater containing 400 mg/L ammonia nitrogen, the system showed a good removal rate and stability in the pollutant removal and phosphorus recovery.

**Table 1 ijerph-19-13950-t001:** MBGS’s treatment of different types of wastewaters and operating parameters.

Wastewater Type	Reactor Type	Initial Characteristics	Volume	Operating Parameters	Removal Efficiency	References
Municipal Wastewater (Synthetic)	glass container	NH_4_-N 99.4 mg/L; TP 13.2 mg/L.; CH_3_COONa 3H_2_O 552.8 mg/L	60 mL	The number of cycles is eight. The first three cycles were 8 h, and the last five cycles were 6 h.	Organic matter 92.69%; NH_4_-N 96.84%; TP 87.16%.	[26]
Municipal Wastewater (Synthetic)	glass container	KH_2_PO_4_ 13.2–21.9 mg/L	60 mL	Six h cycle, light-dark ratio 2:4 h; PPFD illumination is 200 ± 10 μmol/m^2^/s.	Phosphorus 86%	[26]
Simulated wastewater	glass anaerobic bottle	NaAc 113.1 mg/L; NH_4_Cl 30.3 mg/L; K_2_HPO_4_ 50 mg/L;	50 mL	HRT is 12 h.	(COD), NH_4_^+^-N and PO_4_^3−^-P could reach 59.9 ± 6.8%, 78.1 ± 7.9% and 61.5 ± 4.5% on daytime; at night were 47.6 ± 8.0%, 56.5 ± 17.9% and 74.2 ± 7.6%	[51]
Municipal Wastewater (Synthetic)	SBR	COD 320 mg/L; NH_4_-N 35 mg/L; TP 9 mg/L.	6 L	Eight h cycle, water inflow for 3 min, anaerobic for 120 min, aerobic for 210 min, anoxic for 114–142 min, sedimentation for 30 min, and discharge for 3 min. VRT is 50%.	COD 95%; NH_4_-N 97–99%; TP about 93–96%.	[13]
aquaculture wastewater	glass container	COD 280.91 mg/L; NH_4_-N 11.44 mg/L; (NO_3_-N) (NaNO_3_) 16.61 mg/L; NO_2_-N 9.86 mg/L; PO_4_-P 2.83 mg/L	40 mL	Eight h cycle, 36 cycles in total; light intensity: 200 μmol/m^2^/s.	COD 64.8%; NH_4_-N 84.9%; NO_3_-N 70.8%; NO_2_-N 50.0%; P 84.2%	[53]
Domestic sewage (synthetic)	series reactor	COD 300 mg/L; H_4_-N 100 mg/L; PO_4_-P 10 mg/L	1 L	Aeration (0.5 cm/s) and no aeration 60 min: 30 min alternately; HRT is 6 h.	COD 96%; NH_4_-N 99%; TP 50%	[57]
Low Carbon Wastewater (Synthetic)	SBR	PO_4_-P 10 mg/L	0.92 L	Light to dark ratio 12:12 h; HRT is 7.5 h; volume exchange ratio is 53%; 4 h/cycle, 2 min of water inflow, 60 min of no aeration, 173 min of aeration, 2 min of precipitation and 3 min of water effluent.	COD/N = 1, COD 82.5%; TP 98%. COD/N = 2, COD 90.6%; TP 98%.	[58]
Eutrophic Brackish Water (2%, 4%)	SBR	NH_4_-N 50 mg/L; PO_4_-P 10 mg/L	2 L	One hundred and eighty µmol m^−2^ s^−1^ light; HRT is 24 h; VRT is 50%; Ventilation flow 3 L/min; 12 h/cycle, feeding 2 min, no aeration 2 min, aeration 706 min, precipitation 5 min, decantation 3 min, idling 2 min.	2% salinity wastewater, TN and TP 98%; 4% salinity wastewater, TIN 17%.	[59]
Salty wastewater (1–3%)	CFR	COD 600 mg/L; NH_4_-N 50 mg/L; PO_4_-P 10 mg/L	20 L	Three hundred µmol m^−2^ s^−1^ light for 12 h; HRT is 9.5 h Influent flow rate 35 mL/min; The sludge return area is about 4.5 L/min; The aeration zone is about 13.5 L/min.	COD 91.5%; TP 41.8–49.2%; TIN 45.6–55.6%.	[52]

## 4. Biofuels and Bioproducts

Big Environment advocates the establishment of a green and recyclable resource and environmental technology system. The removal of pollutants from wastewater and the realization of resource recovery, has become the developmental trend of wastewater treatment technology [60]. As far as a single MBGS system is concerned, lipids, rich in microalgae, can be used as biofuels, and the microalgae themselves are then processed into biochar [61,62]. Phosphorus and extracellular polymers can be extracted from algal sludge pellets for industrial production [27,63]. The resource recovery of MBGS is shown in Figure 4.

### 4.1. Biofuels

Microalgae can be used to produce biofuels, such as biodiesel, biogas, and biochar. Aerobic granular sludge can also be used for the anaerobic fermentation to produce biogas. Therefore, algal-bacterial symbiotic aerobic granular sludge, has a great potential for biofuel production [64]. Different types of biofuel production from microalgae, as feedstock, are shown in Table 2.

Biodiesel is a renewable energy with a wide range of application values and economic benefits. Biodiesel was originally extracted from edible crops, such as soybean, corn, wheat, and rapeseed, but it cannot be widely used in industrial production, due to its edible value [65]. Later, the biomass was extracted from the plant jatropha to refine diesel oil, but the large planting area and low yield could not be applied to large-scale production [66,67]. Microalgae grow fast, have a high biological yield, are rich in proteins, polysaccharides, lipids, and other substances, and can be produced on a large scale in a small area, making them a potential alternative to fossil fuels [68,69]. Currently, a variety of microalgae with a high lipid productivity show a great potential in diesel production, such as *Chlorella*, *Dunaliella salina*, *Nannochloropsis*, *Phaeodactylum tricornutum*, etc. [68,70].

Meng et al. [71] converted total lipids extracted from algal bacterial sludge, into fatty acid methyl esters (FAMEs), and detected that the main components of FAMEs were pentadecanoic acid, palmitic acid, palmitoleic acid, heptadecanoic acid, and stearic acid. Short-chain fatty acids containing 14–18 carbon atoms, are the main components of biodiesel. Thus, algal granular sludge can be regarded as a potential fuel for diesel production. Liu et al. [72] found that the biodiesel yield of algal sludge pellets was significantly higher than that of aerobic granular sludge. Studies have found that appropriate changes in environmental conditions can promote the lipid production, such as light, light intensity, and influent salinity. Under light conditions, the lipid production was 22.8 mg/g more than under dark conditions [73]. Meng [71] found that the light intensity in the range of 45, 90, 135, 180, and 225 μmol m^−2^ s^−1^ was positively correlated with the lipid production. Both a low salinity (2%) and a high salinity (4%) brackish eutrophication water culture of algal granular sludge can increase the lipid yield from 50.7 mg/g to 64.7 mg/g, 54.1 mg/g to 112.0 mg/g [59]. Meng [52] increased the salinity from 1% to 4%, and the lipid yield gradually increased from 45.9 mg/g to 80.0 mg/g. A high salinity environment favors the lipid production. Commonly used lipid extraction methods from microalgae include near-infrared spectroscopy, electroporation, supercritical fluid extraction, pressurized solvent extraction, organic solvent extraction, and osmotic shock [74]. Cui [27] used pyrolysis to extract the abundant bio-oil and biochar from algal-bacterial sludge granules.

Biogas is produced by the anaerobic digestion of the biomass using anaerobic microorganisms. Anaerobic bacteria in algal sludge particles can be used for the anaerobic digestion to produce biogas. Microalgae have an abundant biomass and can be used to produce biogas. Szwaja [75] found that cyanobacteria-dominated organisms can produce higher methane than other algae. Algal bacterial sludge particles have a great practical value in the biogas production. However, because the microalgae are so small, the cultures are heavily diluted and it is necessary to spend a large amount of energy to recover the biomass, which corresponds to a high percentage (30%) of the total production cost [76]. Although centrifugation is an effective harvesting method, it has a high investment and operating costs. The authors [77] conducted a detailed study of the electrocoagulation, a non-conventional technique for harvesting microalgae, for *Nannochloropsis* sp. They used a current density of 8.3 mA/cm^2^ and obtained the best recovery (>97%) in 10 min without significant changes in the quality of the biomass, in terms of fatty acid and pigment profiles. Microalgal biomass is usually recovered by drying, using ovens, which requires electricity consumption from the grid. To overcome the high energy consumption of this technology, solar drying systems have been developed for agricultural and forest products, although these systems are seasonal. Today, there is a lack of information on the use of solar energy for the production of natural products, mainly algae. However, the use of renewable energy sources is important for the expansion of alternative processes, compared to traditional processes, based on fossil fuel energy. Electrocoagulation and solar drying seem to be potential future trends [78].

**Table 2 ijerph-19-13950-t002:** Different types of biofuel production from microalgae [70].

Microalgae	Algae Type	Biofuel	Productivity of Biofuel	References
*Arthrospira maxima*	Green	Hydrogen, Biodiesel	40–69%	[79,80,81,82,83,84,85,86,87,88,89,90,91,92,93,94,95,96,97,98,99,100]
*Chlamydomonas reinhardtii*	Green	Hydrogen	2.5 mL/h/11.73 g/L	[100,101,102,103,104]
*Chlorella*	Green	Biodiesel		[105,106]
*Chlorella biomass*	Green	Ethanol	22.6 g/L	[107]
*Chlorella minutissima*	Green	Methanol		[108]
*Chlorella protothecoides*	Green	Biodiesel	15.5 g/L	[109,110]
*Chlorella regularis*	Green	Ethanol		[111]
*Chlorella vulgaris*	Green	Ethanol		[112]
*Chlorococcum humicola*	Green	Ethanol	72 g/L or 10 g/L	[113]
*Chlorococcum infusionum*	Green	Ethanol	0.26 g ethanol/g algae	[114]
*Dunaliella* sp.	Green	Ethanol	11.0 mg/g	[115]
*Haematococcus pluvialis*	Red	Biodiesel	420 CJ/ha/yr	[116,117]
*Neochlorosis oleabundans*	Green	Biodiesel	56.0 g/g	[86]
*Platymonas subcordiformis*	Green	Hydrogen		[118]
*Scenedesmus obliquus*	Green	Methanol, Hydrogen		[107,119,120]
*Spirogyra*	Green	Ethanol		[121]
*Spirulina platensis*	Green	Hydrogen	1.8 umol/mg	[122]
*S. platensis* UTEX 1926	Blue-Green	Methane	0.40 m^3^/kg	[123]
*Spirulina* Leb 18	Blue-Green	Methane	0.79 g/L	[124]

### 4.2. Phosphorus

Phosphorus is a key raw material commonly used in industrial production. It is generally obtained by mining phosphate rock. However, phosphorus is also a non-renewable resource and is not able to meet the growing industrial demand. The wastewater contains a large amount of nitrogen and phosphorus and other elements. If the wastewater treatment can be carried out at the same time as the resource recovery, the crisis of resource shortage in the future industrial development process can be greatly reduced. When removing COD and ammonia nitrogen from water, algal sludge granules enrich nutrients, such as nitrogen and phosphorus, in organisms in various forms. Ji [26] studied the mechanism of phosphorus removal from algal-bacteria granular sludge. Studies have shown that 86% of phosphorus can be quickly removed under the reaction conditions of the 6 h light-dark cycle, and phosphorus is assimilated by the microalgae and microorganisms and stored in the organisms in the form of polyphosphate. Phosphate from storage is the main pathway for phosphorus removal, and Pantanalinema is identified as the main P-accumulating algae. The microalgae and biological community of algal-bacterial sludge granules have a good ability to enrich phosphorus, so the extraction of phosphorus from algal-bacterial sludge granules can be a feasible way of phosphorus recovery. The conventional phosphorus recovery method directly inactivates the granules as struvite, which is directly used in agricultural production [125].

### 4.3. N-Acyl Homoserine Lactone (ALE)

Extracellular polymer (EPS) is a natural substance secreted by algal sludge particles, during the formation and resistance to external stimuli. ALE is a polymer that can form hydrogel in EPS, which is easy to regenerate and biodegradable, and can be used as an environmentally friendly industrial material, such as gel and surface coating. ALE is closely related to the stability of algal granular sludge [32]. The algal bacterial granular sludge in the two reactors, was operated under light and dark conditions, and different concentrations of NaCl (10 g/L, 20 g/L, 30 g/L, and 40 g/L) were added at different times. As the salinity increases, the particle stability decreases, the ALE content also decreases from 63.4% and 39.2% to 17.3 and 28.5 mg/g, and the content of the components in ALE that affects the particle gel formation ability and chain flexibility decreases as well [52].

### 4.4. Adsorbents

Algal bacterial sludge particles can produce different surface functional groups, due to their diverse biological communities, and theoretically, the sites formed by these functional groups can bind heavy metals [126,127]. At present, some teams have used algal bacterial sludge for the heavy metal remediation of hexavalent chromium. Yang [128,129,130] used algal-bacteria granular sludge to remove Cr(VI)-containing wastewater for the first time. Studies have shown that the adsorption of hexavalent chromium by particles is highly dependent on the pH value. When the pH is 2, the removal of Cr(VI) is mainly completed by biosorption and bioreduction, and the adsorbed chromium on the particles mostly exists in the form of trivalent, and the Cr(VI) removal rate is as high as 99.3%. Under the conditions of pH 6 and room temperature for 6 h, the adsorption of total chromium by particles can reach 85.1%, and the removal rate is further improved to 93.8% by an external supply of glucose. It is shown that providing a carbon source and natural organic matter can improve the efficiency of the particle removal of Cr(VI). The overall removal rate and stability of algal sludge granules are better than aerobic granules in removing Cr(VI). Yang [129] added a small amount of antibiotics (levofloxacin) and metabolic inhibitors (NaN_3_) to the wastewater, which reduced the removal rate of the aerobic granular sludge by 16.1% or 10.1%, but the removal rate of the algal granular sludge was not significantly affected. Yang [130] added 5 mL/L of Cu^2+^ to the wastewater and reduced Cr(VI) with Cu^2+^, which improved the removal of Cr(VI) by 8.1%, but the total chromium removal was inhibited. An excessive salinity exposure reduces the hexavalent chromium bioremediation capacity [131].

### 4.5. Reduction of the Greenhouse Gas Production

In 2021, the Chinese government issued new environmental protection policies aiming to strengthen the key scientific and technological problems of green and low carbon, and to build a clean, low-carbon, safe, and efficient energy system by 2060. However, the traditional activated sludge technology and aerobic granular sludge often produce a large amount of greenhouse gases, such as CO_2_ and methane, in the process of wastewater treatment, which is not conducive to the development of cleaner production and the establishment of a low-carbon energy system. As a new type of wastewater treatment technology, MBGS can treat wastewater more efficiently. At the same time, it shows a high utilization rate of greenhouse gases, which makes this technology emerge in the process of energy efficiency improvement and carbon emission reduction [132,133,134,135].

Wang [136] ran an algal granular sludge in a closed photo-sequencing batch reactor (PSBR), and CO_2_ and CH_4_ were not detected at the top of the PSRP, which means that the use of the ABGS system can reduce greenhouse gas emissions. It shows that microalgae absorb CO_2_ under photosynthesis. Some microorganisms remove methane through digestion, such as methanotrophs. Ji [137] used the function of microalgae to try to introduce CO_2_ when the algal-bacteria granular sludge was used to treat wastewater. Studies have shown that CO_2_ adds more carbon sources to the system, promotes the accumulation of nutrients, such as nitrogen and phosphorus, and at the same time makes the particles produce more sugar, and the particle structure is more stable. Therefore, an appropriate increase in carbon sources can not only reduce carbon emissions, but also promote particulate the biomass and stability. Safitri [138] inoculated methane-rich activated sludge with cyanobacteria to form an algal-bacteria granular sludge complex. During the stable operation of the reactor, the removal rate of dissolved methane reached 84.8%. Methanophiles are the main group of organisms that remove dissolved methane. There are a variety of microalgae and microbial communities in the algal sludge particles, which absorb carbon sources well and show great potential in the realization of negative carbon technology in wastewater treatment.

## 5. Challenging Prospects

Reactor Design. Existing MBGS experiments are run on small benchtop scales, lacking the large-scale practical application experience [139]. For example, sequencing batch reactors (SBRs) are commonly used to culture particles, and SBRs and continuous flow (CFR) experiments simulate wastewater treatment. Meng [52] treated saline wastewater with algal bacterial granular sludge in a continuous flow reactor, and the system maintained a good stability and a high nitrogen and phosphorus removal efficiency. At present, the CAS process is commonly used in sewage plants to treat industrial wastewater, and the CAS process and equipment cannot be applied to MBGS. Therefore, it is expected to design a wastewater treatment reactor, in the future, to enable MBGS to operate on a large scale and for a long period of time.

Granulation and particle stability. Research on the MBGS granulation and its structural stability has made some progress. It has been shown that at the macroscopic scale, the structural morphology is influenced by external environmental conditions, such as reactors and reaction systems, while at the microscopic scale, the structural morphology is regulated by microorganisms and sludge physicochemical properties. However, it should be noted that the rapid formation of MBGS with a stable structure and the maintenance of its long-term structural stability, in practical applications, avoiding problems such as particle disintegration, sludge floating, and degradation of the decontamination performance, will be the key to the industrial application of the MBGS technology. Regarding key issues, such as system granulation and a long start-up time, many teams have tried various methods to shorten the start-up time, such as inoculating mature algal bacterial sludge, adding metal ions, adding condensation nuclei, and using inter-microbial signal molecules. Gikonyo [140] inoculated oxygen-containing light particles in the activated sludge for hydrodynamic granulation. Zhang [141] added the signal molecule AHL to the reactor during the granulation stage. However, the granulation mechanism of MBGS remains to be explored. At present, the undesirable characteristics of microalgae may affect pelleting, but they have rarely been studied, and this is a task that should not be neglected for the study of MBGS. In the near future, we can focus on the improvement of signal molecule detection methods, an in-depth study of the distribution pattern and mechanism of signal molecules, and study the coupling between the structural and morphological characteristics of MBGS and the distribution pattern of the microbial community evolution, with the help of modern molecular biotechnology and other advanced in situ characterization techniques, to further understand the formation process and structural stability of MBGS at the microscopic scale, and to construct a model of the MBGS granulation process. The model of the MBGS granulation process can be developed to propose the best operating control conditions for rapid granulation and the long-term structural stability of MBGS, and accelerate the industrial application of the MBGS technology.

Aeration. The rate of aeration affects the intensity of shear, gas balance, and substrate diffusion, leading to different properties of the microalgal-bacterial consortium [142]. The gas balance between O_2_ and CO_2_ may further play an important role in the algal metabolism [143]. Microalgal-bacterial assemblages have the potential to self-sustain their gas usage, by introducing O_2_ from microalgae to bacteria and supplying CO_2_ in the opposite way [144]. Theoretically, the MBGS process requires little external aeration, but remains a problem to overcome in different situations. Indeed, if other factors, such as light and nutrition, do not limit the growth of each member, gases can circulate within the system without the need for external aeration. However, a number of factors, including changes in growth status, metabolic inefficiencies, and the resistance to gas transfer, may make the consortium undesirable [145]. In particular, the niche of and challenges of zero aeration, if desired, include the need for optimized process parameters, potential solutions to overcome nutrient limitation [146], and enhanced gas exchange between microalgae and bacteria at the organism and reactor levels, to achieve the MBGS systems with a minimal gas input [147].

Economic recovery of resources. The algal-bacterial consortium, not only effectively removes nutrients from wastewater and converts them into biomass, but also recovers nutrients or carbon sources from the wastewater. The consortium converts phosphorus from wastewater into polyphosphates, which can be used in medical, biomaterials, and food applications, due to their antiseptic, cytoprotective, and antiviral activities [148]. For example, polyphosphates from algae can inhibit the HIV-1 virus. Calcium polyphosphates can be used for tissue regeneration. Polyphosphates can also be used in the food industry, such as ham and bacon. The carbon source in algal cells is converted into the form of polysaccharides and biofacts. Majee et al., 2017 [149] summarized the properties of polysaccharides produced by several marine bacteria and algae and their applications in pharmacology, diagnostics, and cosmetics, reflecting their sustainable use as recycled resources. The consortium has a very strong nitrogen recovery efficiency. Luo et al., 2019 [150] concluded that the nitrogen-fixing bacteria and microalgae consortium can be used directly as organic fertilizer and the protein of the consortium can be used as a substitute for fish feed, the main component of this substitute being fish meal protein. However, the use of biomass in food, medicine, and agriculture, is subject to many limitations because microalgae can remove heavy metals that are hazardous to human health or cause secondary environmental pollution. Therefore, further removal of heavy metals or recalcitrant compounds from economically recoverable resources will be a key breakthrough in biomass utilization research. In addition, the conversion of this biomass into energy through the anaerobic digestion of biogas can prevent direct contact of this biomass with the human body.

## 6. Conclusions

The shift from wastewater treatment to resource conversion and recycling has become a trend towards environmentally sustainable strategies. This review highlighted the research progress of algal-bacterial symbiotic aerobic sludge particles in wastewater treatment and discussed the factors affecting particle formation, process parameters, resource recovery and reuse capacity, and carbon emission reduction. Clearly, MBGS shows a great potential in clean production and environmental sustainability. However, the reactor design for a large-scale operation, a long granulation time, and unstable long-term operation stability is still the bottleneck of this technology. In addition, MBGS needs to provide some aeration in both the start-up and operation phases. Thus, it would be a breakthrough if zero aeration is needed while wastewater is being treated by MBGS technology.

## Figures and Tables

**Figure 1 ijerph-19-13950-f001:**
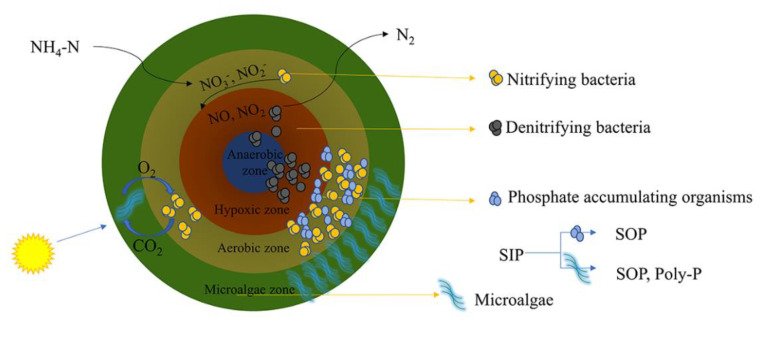
MBGS structure diagram.

**Figure 2 ijerph-19-13950-f002:**
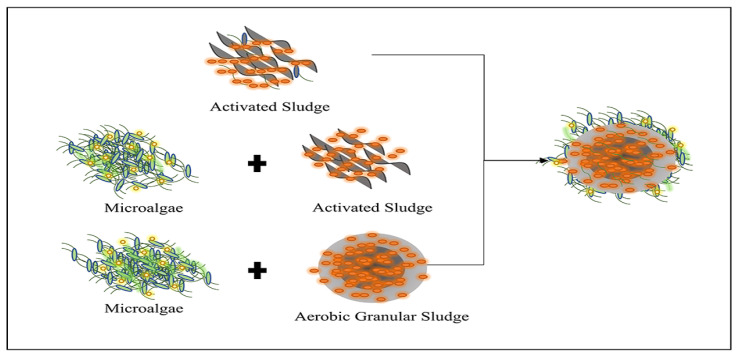
MBGS formation strategy.

**Figure 3 ijerph-19-13950-f003:**
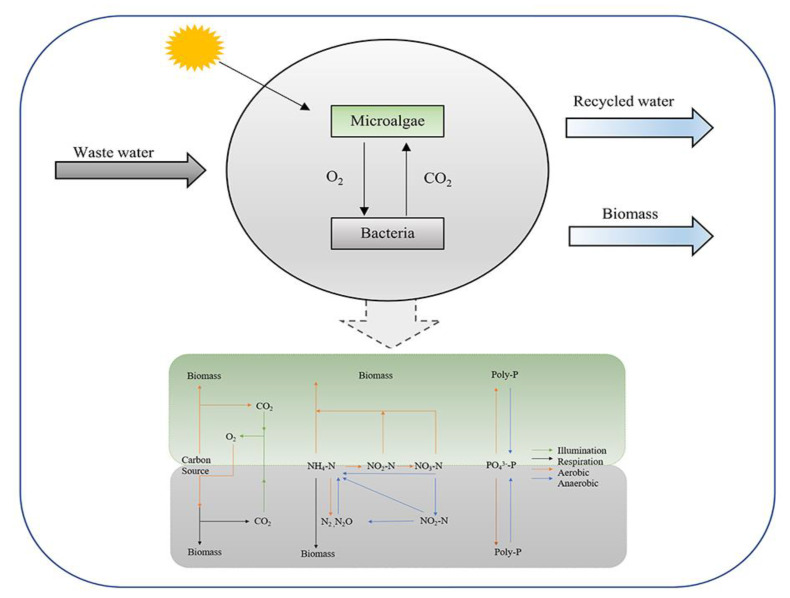
The principle of the MBGS treatment of wastewater.

**Figure 4 ijerph-19-13950-f004:**
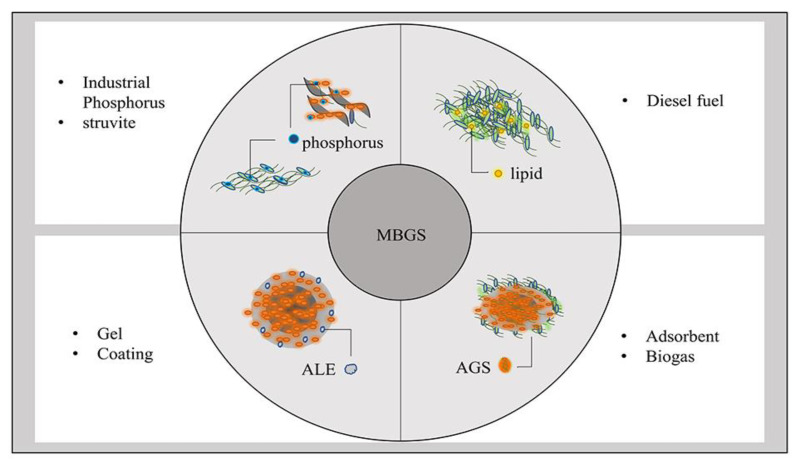
Recycled resources from MBGS.

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
