# Peer review of "Current Progress, Challenges and Perspectives in the Microalgal-Bacterial Aerobic Granular Sludge Process: A Review"

_ijerph, 2022, doi:10.3390/ijerph192113950_

Round 1

Reviewer 1 Report

This research is  important as a review in the field of water purification, and I recommend publishing the work after adding some results

Author Response

Thank you for offering us an opportunity to improve the quality of our submitted manuscript (ijerph-1965878). We appreciated very much the reviews’ constructive and insightful comments. In this revision, we have addressed all of these comments and suggestions. We hope the revised manuscript has now met the publication standard of your journal.

On the next pages, our point-to-point responses to the queries raised by the reviewer are listed.

Comment 1: This research is  important as a review in the field of water purification, and I recommend publishing the work after adding some results.

Response: We have added some results to the revised manuscript. For example, we added the paragraph " Thus, it would be a breakthrough  if zero aeration is needed whilewastewater is being treated by MBGS technology." to the conclusion of the article

Special thanks to you for your good comments and suggestions.

Reviewer 2 Report

Please find my comments for Authors in the attached file.

Author Response

Thank you for offering us an opportunity to improve the quality of our submitted manuscript (ijerph-1965878). We appreciated very much the reviews’ constructive and insightful comments. In this revision, we have addressed all of these comments and suggestions. We hope the revised manuscript has now met the publication standard of your journal.

On the next pages, our point-to-point responses to the queries raised by the reviewer are listed.

Comment 1: You need to add an extra paragraph in the abstract. Kindly focus on presenting the key progress and novelties of MBGS derived from the review itself.

Response: We added the key progress and novelties of MBGS in the abstract of the revised manuscript, as our abstract states” The information in this review will help us better understand current progress  and future direction of the MBGS technology development. It is expected that this review will provide  a sound theoretical basis  for the practical applications of MBGS technology in environmentally sustainable wastewater treatment, resource recovery, and system optimization.”

Comment 2: The novelty of this literature research should be inserted in the text clearly.

Response: We inserted the novelty of the literature study in the introduction of the revised manuscript, and we inserted " MBGS is a composite bioconcentration technology based on the coupling of microalgae and sludge. MBGS not only combines high biomass and high treatment efficiency of sludge with high added value and resource recovery capability of microalgae but also solves the problem of poor settling performance and difficulty in recovery of microalgae in the water treatment process. At the same time, microalgae in situ oxygen production and bacteria form small bioconcentration of oxygen and carbon dioxide cycle, which can use the light source as the only energy source to replace external aeration with microalgae oxygen production under light conditions. This makes MBGS promising as a new green and sustainable new process to achieve carbon neutrality goals in future municipal wastewater treatment. At this stage, MBGS is experimented with treating various industrial wastewater, such as municipal wastewater, farm wastewater, and saline wastewater, and has shown notable removal capacity." in the second paragraph of the introduction.

Comment 3: The regeneration cost and process of the MBGS after treatment should be evaluated.

Response: We added the cost and process of regeneration after MBGS treatment in the revised manuscript. We added the section on economic recovery of resources to the Challenging prospects of the article.

Comment 4: Table 1 lacks initial characteristics of wastewaters (e.g. COD, TP, TN, …). This

information is incomplete without those data.

Response: We added the initial characteristics of the wastewater in Table 1 in the revised manuscript.

Comment 5: The stability of the MBGS should be presented deeply.

Response: The stability of MBGS has been described in more depth in our revised manuscript , for example, we wrote " Research on MBGS granulation and its structural stability has made some progress. It has been shown that at the macroscopic scale, the structural morphology is influenced by external environmental conditions such as reactor and reaction system, while at the microscopic scale, the structural morphology is regulated by microorganisms and sludge physicochemical properties. However, it should be noted that the rapid formation of MBGS with stable structure and the maintenance of its long-term structural stability in practical applications, avoiding problems such as particle disintegration, sludge floating and degradation of decontamination performance, will be the key to the industrial application of MBGS technology. "and" In the near future, we can focus on the improvement of signal molecule detection methods, in-depth study of the distribution pattern and mechanism of signal molecules, and study the coupling between the structural and morphological characteristics of MBGS and the distribution pattern of microbial community evolution with the help of modern molecular biotechnology and other advanced in situ characterization techniques, to further understand the formation process and structural stability of MBGS at the microscopic scale, and to construct a model of MBGS granulation process. The model of MBGS granulation process can be developed to propose the best operating control conditions for rapid granulation and long-term structural stability of MBGS, and accelerate the industrial application of MBGS technology.” in the Challenging prospects of the article.

Comment 6: The limits of this biological treatment technology especially for industrial wastewater should be well discussed.

Response: The limitations of this biological treatment technology for industrial wastewater are discussed in more depth in our revised manuscript. We wrote in the introduction that " The limitations of MBGS for industrial wastewater treatment are mainly the need to combine physical and biochemical means; secondly, the generation of greenhouse gases such as methane; and the poor resource recovery performance."

Comment 7: The formation mechanism of MBGS might provide a new insight for the rapid

granulation. It is suggested to state the main mechanisms in the paper.

Response: The mechanism of MBGS formation provides new insights into rapid granulation that we have described in the revised manuscript. In the second part of our article, we wrote "The formation mechanism of MBGS may provide new insights into rapid pelletizing. For example, rational control of the selection pressure; helps to control the growth of biomass, improve the hydrophobicity of the pellets, and accelerate pelletizing".

Comment 8: Generally, some undesirable properties of microalgal cells, namely small cell size, poor settling ability and strong electrostatic repulsion make it challenging to form the MBGS.

Response: We added the effect of undesirable properties of microalgal cells on pelletization to the revised manuscript. We wrote "At present, the undesirable characteristics of microalgae themselves may affect pelleting, but they have rarely been studied, and this is a task that should not be neglected for the study of MBGS." in the Challenging prospects of the article.

Comment 9: The names of all species should be in italics.

Response: In the revised manuscript we changed the names of all species to italics.

Special thanks to you for your good comments and suggestions.

Reviewer 3 Report

The length of a review paper is too short. The authors have referred very less reference research articles.

The authors should expand their studies.

Only one table is available.

The manuscript does not provide scientifically sound.

The introduction is too common.

The results and discussion part is too short.

Where are the material and method parts?

The author should include the research methodology. 

Author Response

Thank you for offering us an opportunity to improve the quality of our submitted manuscript (ijerph-1965878). We appreciated very much the reviews’ constructive and insightful comments. In this revision, we have addressed all of these comments and suggestions. We hope the revised manuscript has now met the publication standard of your journal.

On the next pages, our point-to-point responses to the queries raised by the reviewer are listed.

Comment 1: The length of a review paper is too short. The authors have referred very less reference research articles. The authors should expand their studies.

Response: We have added content and references to the articles in the revised manuscript and expanded their research, allowing for an increase in the length of the articles.

Comment 2: Only one table is available.

Response: We have added a table to the revised manuscript which describes the production of different types of biofuels from different microalgae.

Comment 3: The manuscript does not provide scientifically sound.

Response: We examined the manuscript and determined that it was scientifically sound. We have added a lot of scientific basis to the revised manuscript.

Comment 4: The introduction is too common.

Response: We have enriched and refined the introduction in the revised manuscript. For example, we have added in the introduction the novelty of the literature study and the limitations of this technology especially for industrial wastewater.

Comment 5: The results and discussion part is too short.

Response: We added the Results and Discussion section to the revised manuscript, allowing for an increase in the length of the section. For example, we added the paragraph " Thus, it would be a breakthrough  if zero aeration is needed whilewastewater is being treated by MBGS technology."

Comment 6: Where are the material and method parts?

Response: What we learned before is that the general review articles do not have a materials and methods section.

Comment 7: The author should include the research methodology. 

Response: What we learned before is that the typical review article does not have a section on research methods.

Special thanks to you for your good comments and suggestions.

Round 2

Reviewer 3 Report

The authors haven't revised the manuscript as per the reviewer's comments.